

# Influence of parental involvement and parenting styles in children's active lifestyle: a systematic review

Marta Vega-Díaz[1], Higinio González-García[2] and Carmen de Labra[3]

[1] Faculty of Nursing and Podiatry, University of A Coruña (UDC), A Coruña, Galicia, Spain
[2] Education, Universidad Internacional de La Rioja (UNIR), TECNODEF Research Group, La Rioja, Spain
[3] 3NEUROcom, Center for Advanced Scientific Research (CICA), Biomedical Research Institute of A Coruña (INIBIC), Faculty of Nursing and Podiatry, University of A Coruña, A Coruña, Spain

Corresponding author
Marta Vega-Díaz,
marta.vega.diaz@udc.es

## ABSTRACT

**Background:** Parents influence their children's lifestyles through modeling and support, which modifies how children approach physical activity. As such, this systematic review aims to know the influence of parental involvement in children's active lifestyles and the influence of parenting styles on children's practice and motivation towards physical activity.

**Methodology:** PubMed, Google Scholar, Scopus, ResearchGate, and Web of Science databases were searched from 15 July 2022 to 30 August 2022. The publication date of the included manuscripts was between November 2012 and January 2021. The methodological quality of the studies was analyzed using the Scale for Evaluating Scientific Articles in Social and Human Science. Furthermore, it was utilized the Preferred Reporting Items for Systematic Reviews and Meta-Analyses 2020.

**Results:** The sample of the 10 included studies (in which different self-report measures were administered) was 1,957 children and their parents. In one study, parental involvement across limits decreased sedentary behaviours. In another, parent-child co-participation in physical activity improved participation in it. In one study, democratic parents predicted moderate-vigorous physical activity. In another, authoritarian styles were associated with sports practice. In another, permissive styles were associated with the worst physical activity practice. In one study, permissive parents were not significantly related to physical activity. In another, negligent parents were negatively associated with sports practice. In two studies, parenting styles were not associated with physical activity. In a study, the autonomy support of permissive parents and the structure of democratic parents is related to autonomous motivations. The coercive control of authoritarian parents and the lack of structure of negligent parents are related to non-self-determined motivations or amotivation.

**Conclusions:** Parental involvement contributes to children's participation in physical activity. There is no unanimity in the results obtained in parenting styles and the practice of physical activity. Democratic and permissive styles are associated with self-determined motivations, as opposed to negligent and authoritarian ones. The results obtained have been extracted from studies where different self-report measures are administered, so it would be advisable to continue researching this subject.

# INTRODUCTION

Inadequate physical activity (PA) is a worldwide public health problem and one of the main risk factors for premature mortality (*Lee et al., 2012*; *Machado-de-Recende et al., 2019*). At least 3.2 million people die each year for not being able to join a program of 30 min of PA per day/several days a week (*World Health Organization, 2020*). The 2011-2012 National Health Survey indicated that, in Spain, 40% of the adult population and 12.1% of the children population declare themselves sedentary (*National Health Survey of the Government of Spain, 2014*). More specifically, 8.2% of boys are basically inactive, the percentage rising to 16.3% of girls (*Díez, 2017*). These data are similar to those presented by the *World Health Organization (2016)* which estimated that, worldwide, only 19.3% of children and adolescents were active. The situation described is worrying because it is known that involvement in physical sports activities decreases from childhood to adolescence and over the years (*World Health Organization, 2016*). The low levels of PA have caused an increase in the incidence of several diseases, such as obesity, cancer, metabolic syndrome and cardiovascular diseases, among others (*Guo et al., 2019*; *Lätt et al., 2015*). Considering that these diseases have a high impact on the use of health services and the consequent medical expense, it seems important to try to modify the sedentary lifestyle from an early stage of life. In particular, it is known that being active while childhood increases the chances of being active during adulthood (*Biddle et al., 2010*; *Trudeau, Laurencelle & Shephard, 2004*).

However, previous systematic reviews report on parent-focused intervention programs to improve children's school PA (*Kovács et al., 2022*) and PA in general (*Gatus et al., 2022*). Systematic reviews also explore parents' influence on different types and intensities of PA (*Edwardson & Gorely, 2010*). The few systematic reviews that examine parenting styles (PS) and PA practice focused on samples from the United States with children under 2 to 12 years old (*Lindsay et al., 2018*). This work aims to obtain a more general vision of parents' role (through parental involvement and PS) in active lifestyles and PA, adding a fundamental variable for them: motivation. Therefore, this work will provide scarcely explored information, offering a current perspective on the importance of parental involvement and PS in active lifestyles and motivation towards physical activity, something fundamental for human health.

Parental involvement is defined as a set of actions, beliefs and signs of implication that parents show to their children (*Desimone, 1999*). Parental involvement in children's active lifestyles has been examined in some previous work (*Glozah, Oppong & Kugbey, 2018*; *Hendrie et al., 2012*; *Rebold et al., 2015*). The active lifestyle is understood as the practice of regular PA, of moderate to vigorous intensity, for a minimum of 30 min daily, this time being able to obtain from leisure activities and free time, work or household chores (*Sañudo, Martínez & Muñoa, 2012*). Parental involvement includes behaviors such as providing values to children, talking with them (*Scribner, Young & Pedroza, 1999*), giving resources, investing time with them, and money (*Bower & Griffin, 2011*). In the PA field,

parental involvement can be done directly or indirectly. In direct parental involvement, parents perform PA together with their children, paying the fees for them to participate in physical activities, talking with them about the benefits of PA (*Beets, Cardinal & Alderman, 2010*), and acting as role models (*Beets, Cardinal & Alderman, 2010*; *Ormrod, 1999*). Considering the modeling of parents individually, *Nuviala, Ruíz & García (2003)* found that children practiced more sports when the mother was the PA practitioner. *García-Ferrando (1993)* determined the importance of parental modeling, since 76% of the children whose mothers practiced sports were active. However, only 49% of young people were active if inactive mothers raised them. Regarding indirect parental involvement, parents behave in a passive way that does not require them to participate in their children's PA personally. However, they may do so by watching children during sports competitions and training (*Hendrie et al., 2012*; *Hingle et al., 2010*).

Some previous works show that the PA of children is influenced by PS (*Jago et al., 2011*; *Sekot, 2019*; *Van-der-Geest et al., 2017*). PS can be defined as the attitudes that parents express when living with their children and the emotional climate that exists between them (*Darling & Steinberg, 1993*). PS are exercised through support, control, parental demands, communication (*Baumrind, 1967*), affection, autonomy (*Schaefer, 1959*), psychological control and behavioral control (*Barber, Olsen & Shagle, 1994*), and restriction (*Becker, 1964*), *etc.* Considering the different variables that can be found within the PS, different types of them can be distinguished. These styles are democratic, authoritarian, permissive, and negligent. Generally, democratic parents are demanding and receptive. Authoritarian styles are demanding and directive but not receptive. Permissive parents are the most receptive, but they are not demanding. Neglectful styles are non-demanding, non-directive, and non-receptive. It has been found that specific behaviors that parents adopt during nurture, such as over-demands are negatively related to the practice of PA (*Jago et al., 2011*). Likewise, when PS are governed by restrictions, rigid discipline, and over-exertion, children do not report good levels of PA practice. In other words, there are situations where children do not participate in physical activities because their parents do not permit them to enroll (*Nuviala, Ruíz & García, 2003*). On other occasions, rigid discipline and overexertion (*Dencker et al., 2006*) suppress the children's PA practice (*Pelegrín, González-García & Garcés-De-Los-Fayos, 2019*). At the opposite extreme, parents who use reasoning and accept their children's requests facilitate their participation in PA (*Hennessy et al., 2010*). In the same way, parents who support their children in practicing PA improve their participation levels (*Hennessy et al., 2010*; *Jago et al., 2011*).

PS also exert their role in the motivation towards the practice of PA (*Keegan et al., 2009*; *Kitzman-Ulrich et al., 2010*; *Tang et al., 2018*). Motivation is a variable that promotes the onset or maintenance of an action or behavior (*Moral-García, Uchaga-Litago & Ramos-Morcillo, 2020*). Motivation can appear as a response to exogenous factors (non-self-determined motivation), or people can act of their own volition (self-determined motivation) (*Ponseti et al., 2019*). Non-self-determined motivation can evolve into self-determined motivation if people perceive the activity they carry out as part of their value (*Laukkanen, Sääkslahti & Aunola, 2020*). In general, democratic styles are negatively related to not self-determined motivation (*Tang et al., 2018*). Therefore, children will
participate in PA through their motives (*Ryan & Deci, 2000*). Authoritarian and permissive styles are negatively related to self-determined motivation (*Tang et al., 2018*). In these cases, the children will participate in the PA to obey their parents (authoritarian) or obtain rewards (permissive). Neglectful parents are positively linked to non-self-determined motivations (*Rubin, 2017*). In this situation, the children will not be inherently motivated to participate in PA (*Ryan & Deci, 2000*).

Literature has been mainly focused on examining, separately, the role of parents' involvement in active lifestyles (*García-Ferrando, 1993*; *Lev et al., 2020*), or PS in the practice of PA (*Jago et al., 2011*; *Lee & Román, 2014*; *Martínez-López et al., 2014*) or PS in the motivation towards PA (*Huffman et al., 2018*; *Pendlebury et al., 2013*). This systematic review goes a step further, aiming not only the study of all the variables together (active lifestyles regarding parental involvement and the practice and motivation toward PA regarding PS), but also to provide information about the parental variables that act negatively and positively on the adoption of active lifestyles of the children.

## SURVEY METHODOLOGY

### Method

A systematic review was carried out according to the Preferred Reporting Items for Systematic Reviews and Meta-Analyses (*PRISMA, 2020*) statement (*Page et al., 2021*). In addition, the review was registered in OSF (https://osf.io/a73yd/) and PRISMA-2020 checklist can be found in Appendix I.

### Search strategy

Potential studies were identified by combined search processes, clearly planned and ordered. First, the PubMed, Google-Scholar, ResearchGate, Scopus, and Web of Science databases were consulted, with the following search terms included in Boolean search strategies: (parents involvement) "AND" (active lifestyles); (parenting styles) "AND" (motivation); (parenting styles) "AND" (physical activity). By using filter criteria of the respective databases, the search was limited to publication dates (from 1 January 2012 to 31 July 2022), and searches were in English language. Recent manuscripts (last decade) have been included because PS vary over time, and we wanted to obtain results from today's society. The change of PS over time has been supported by *Ching-Man, Wai-Man & Sin-Min (2019)*. Considering the above, being aware of the cultural changes in society in the last decade (*e.g.*, immigration, parents' role, family structure, *etc.*) (*National Institute of Statistics, 2023*; *Zervides & Knowles, 2007*) it is convenient to investigate PS from a recent perspective.

### Study selection

The articles included in the present systematic review were: those that examined the role of parental involvement in the active lifestyles of children, those that explored the connection between PS (classified as democratic, permissive, authoritarian, and neglectful) with the practice of light, moderate-vigorous PA, *etc*. In addition, the articles that examined the connection between the aforementioned PS and the self-determined or non-self-

determined motivation toward PA, those that were published in a peer-reviewed journal and written in English.

On the other hand, literature reviews, abstracts, editorial comments, and letters to the editors were excluded. It was decided to exclude systematic reviews to include original studies in which information on the study's design (longitudinal or cross-sectional), the context, and the geographical location of the study could be clearly observed. For instance, all systematic reviews do not detail some sample characteristics (only fathers, only mothers, fathers, and children), their age, and other descriptive information. Moreover, including systematic reviews may add some biases related to the treatment of the information.

After the search process, 996 manuscripts were obtained from the different databases. Likewise, 55 registers were obtained. Then, 1,051 manuscripts were obtained from database and registers index. On the other hand, through other methods, three records of interest were registered (one record was identified through websites, and two through organizations). Of the 1,051 manuscripts identified *via* databases and registers, nine were eliminated in this first recovery phase for being duplicates. Therefore, 1,042 were screened. Of the 1,042 manuscripts, 1,011 were excluded during the abstract and title screening step. At this stage, it was not considered necessary to recover any manuscripts. Therefore, 31 manuscripts moved to the full-text screening step phase. Of these 31 manuscripts, 21 were eliminated for the following reasons: two because they were reviews, 18 because they were duplicates, and one because they did not contain all the necessary data that has been detailed in the inclusion criteria. Regarding studies obtained by other methods, three reports were assessed for eligibility. However, all three were eliminated for not rigorously meeting the inclusion criteria of the systematic review. Therefore, finally 10 articles were included in the systematic review. The article selection process can be found in Fig. 1.

## Quality assessment

Two reviewers independently assessed the quality of the included studies using the Scale for Evaluating Scientific Articles in Social and Human Science (SSAHS; *López-López, Tobón & Juárez-Hernández, 2019*). All included studies were scored according to eight specific criteria (cover and summary, introduction, methodology, results, discussion, references, appendices, styles and- format) that were derived from items 1 to 21 of the aforementioned scale. The SSAHS scale was considered a suitable starting point to assess the quality of observational studies. This 21-item scale guides the critical evaluation of scientific articles in Social and Human Sciences. It was utilized the Preferred Reporting Items for Systematic Reviews and Meta-Analyses, 2020 (*PRISMA, 2020*). The authors followed the guideline items ensuring that the systematic review is transparent and accurate. To do this, it was indicated why the review was carried out, what was done (how the studies were identified and selected) and what was found. Regarding the quality of the study reports, the mean score obtained with the Scale to Evaluate Scientific Articles in Social and Human Sciences (SSAHS) (*López-López, Tobón & Juárez-Hernández, 2019*) was 3.77 (minimum: 1, maximum: 5). Specifically, the levels are: 1 = very low level; 2 = low level; 3 = medium level; 4 = medium-high level; and 5 = very high level. This means that

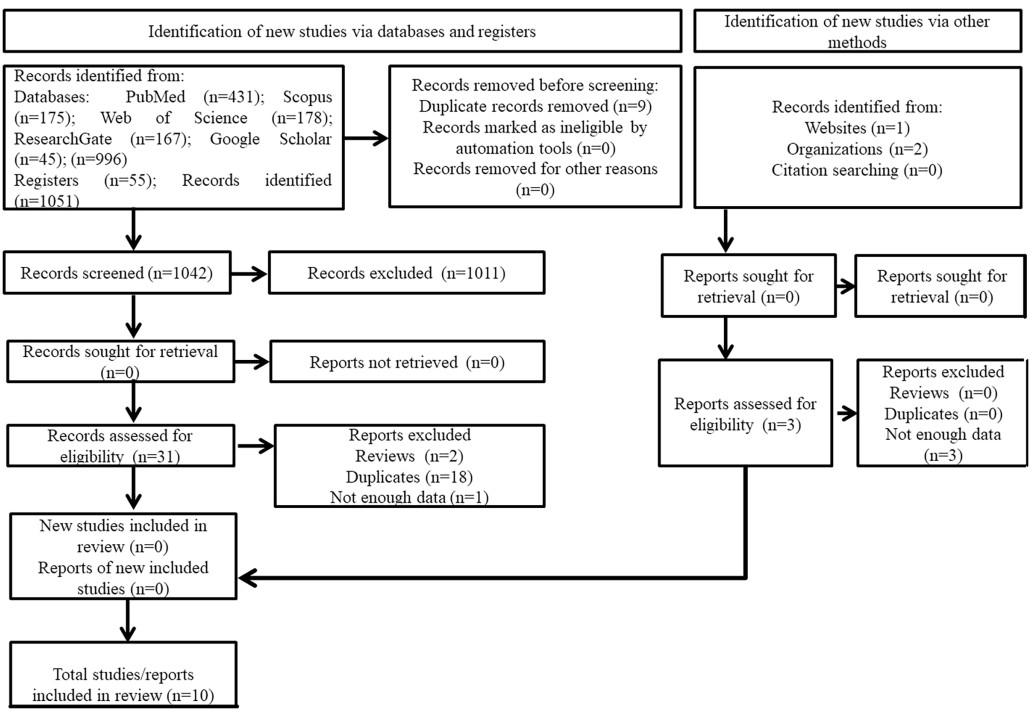

**Figure 1 Flow chart of the studies selection in the systematic review.**

the quality of the articles included in this work is close to a medium-high level (since it is close to four points). Regarding the quality of this systematic review, it has obtained a score of 4.38 (between the medium-high and very-high levels) (*López-López, Tobón & Juárez-Hernández, 2019*). The detailed data are presented in Table 1.

## RESULTS

Table 2 summarizes the main characteristics of each study. Four studies were held in the USA, one in Australia, one in Asia, two in Europe, and two in Israel. The total size of the sample was 1,957 children and their respective parental figures. One of the ten studies exclusively included parental-men figures. Five of the ten studies included both parents and children. Four of the ten studies exclusively included children.

If you wish to obtain more information about the measures used to examine parental involvement, PS, and PA practice, it is suggested that you consult Table 2. It details those used in each of the 10 included studies.

The detailed data are presented in Table 2. Figure 2 and Table 3 summarize the results obtained. In general, direct parental involvement through modeling (*Alia et al., 2013*; *Sterret et al., 2013*), the participation of parents and children together in sports (*Lev et al., 2020*; *Yazeedi et al., 2021*), and the indirect participation in PA (advice sheets making or newsletters) (*Verhees et al., 2020*) may promote active lifestyles. It is usual that when PS raises children with parental support (*Yaffe, 2018*) and affection (*Huffman et al., 2018*), adequate levels of PA practice are registered. However, in two of the ten works, the result was the opposite (*Langer et al., 2014*; *Saunders et al., 2012*). On the other hand, support

**Table 1** Analysis of the selected studies' methodological quality (*n* = 10).

| Study | 1 | 2 | 3 | 4 | 5 | 6 | 7 | 8 | 9 | 10 | 11 | 12 | 13 | 14 | 15 | 16 | 17 | 18 | 19 | 20 | 21 | Total score for each study | Mean of the 21 items for each study |
|---|---|---|---|---|---|---|---|---|---|---|---|---|---|---|---|---|---|---|---|---|---|---|---|
| *Alia et al. (2013)* | 4 | 5 | 3 | 4 | 5 | 3 | 4 | 3 | 5 | 5 | 3 | 5 | 5 | 5 | 5 | 4 | 3 | 3 | 1 | 3 | 5 | 83 | 3.95 |
| *Huffman et al. (2018)* | 4 | 3 | 3 | 3 | 5 | 4 | 3 | 2 | 5 | 5 | 3 | 5 | 5 | 5 | 5 | 4 | 5 | 1 | 1 | 3 | 5 | 79 | 3.76 |
| *Langer et al. (2014)* | 4 | 4 | 4 | 4 | 5 | 3 | 3 | 1 | 4 | 5 | 4 | 5 | 5 | 5 | 3 | 5 | 4 | 2 | 1 | 4 | 5 | 80 | 3.80 |
| *Lev et al. (2020)* | 4 | 4 | 4 | 3 | 5 | 3 | 4 | 1 | 5 | 3 | 4 | 5 | 5 | 5 | 3 | 4 | 3 | 3 | 1 | 3 | 5 | 77 | 3.66 |
| *Laukkanen, Sääkslahti & Aunola (2020)* | 4 | 3 | 4 | 2 | 5 | 4 | 4 | 3 | 4 | 2 | 4 | 5 | 5 | 5 | 3 | 4 | 2 | 2 | 1 | 2 | 5 | 73 | 3.47 |
| *Saunders et al. (2012)* | 4 | 3 | 3 | 3 | 5 | 4 | 3 | 2 | 4 | 5 | 3 | 5 | 5 | 5 | 3 | 2 | 1 | 1 | 1 | 3 | 5 | 70 | 3.33 |
| *Sterret et al. (2013)* | 4 | 4 | 4 | 4 | 5 | 5 | 4 | 3 | 5 | 5 | 5 | 5 | 5 | 5 | 5 | 4 | 4 | 3 | 1 | 4 | 5 | 89 | 4.23 |
| *Verhees et al. (2020)* | 4 | 3 | 4 | 4 | 5 | 4 | 3 | 3 | 4 | 5 | 4 | 5 | 5 | 5 | 4 | 4 | 3 | 1 | 4 | 3 | 5 | 82 | 3.90 |
| *Yaffe (2018)* | 4 | 4 | 4 | 4 | 5 | 4 | 4 | 3 | 5 | 5 | 5 | 5 | 5 | 4 | 5 | 2 | 3 | 1 | 1 | 4 | 5 | 82 | 3.90 |
| *Yazeedi et al. (2021)* | 3 | 3 | 3 | 2 | 5 | 4 | 4 | 3 | 4 | 5 | 5 | 5 | 5 | 5 | 3 | 3 | 4 | 3 | 1 | 3 | 5 | 78 | 3.71 |
| Study | | | | | | | | | | | | | | | | | | | | | | | |
| Present systematic review | **4** | **5** | **4** | **4** | **5** | **4** | **4** | **3** | **4** | **5** | **4** | **5** | **5** | **5** | **3** | **5** | **4** | **5** | **4** | **5** | **5** | **88** | **4.38** |

**Notes:**

The numbers of the columns corresponded to the 21 items of the EACSH scale.

The study of this systematic review is not included within the 10 articles examined in this research, which is why it is specified (*n* = 10)

Items 1 to 4 evaluate the cover and summary. Items 5 to 7 examine the introduction. Items from 8 to 11 the methodology. Items 12 to 14 the results. Items from 15 to 17 the discussion. Item 18 the references. Item 19 the appendices. Items 20 and 21 style and format.

The quality of this systematic review is shown in bold.

1 = very low level; 2 = low level; 3 = medium level; and 4 = medium-high level; and 5 = very high level.

1. The title describes the essential topic of the article, it is simple and clear, attractive and brief, it has less than 16 words, it is centered and the words of more than four letters begin with a capital letter.
2. The names of the authors are indicated after the title, the surnames are linked with a hyphen (unless it is a single surname), the institutional affiliation and the email of each author are added.
3. The abstract is in Spanish and English and is presented with a maximum of 250 words (or the number of words allowed by the journal in which it is expected to be published). In addition, it briefly describes the problem, objective, methodology, main results and conclusions of the study.
4. The number of keywords is between 4 and 8 (or within the range that allows the journal in which it is expected to publish), were extracted from a thesaurus of the discipline, are written in lower case, separated by a comma and in alphabetical order.
5. It begins with some attractive paragraphs that present the relevance, justification, and context of the topic, focusing the reader on the problem of study and motivating them to read the article.
6. A critical review is made of the main studies carried out on the problem, considering the purposes of the study, from the most general to the particular. Furthermore, it is based on paraphrased quotes in most cases, integrating recent information with historical information.
7. The objectives are relevant, they are related to the problem of exposed research and are clearly written (they have a verb in the infinitive, object, subject and context of the study).
8. The type of study carried out is described, for example, if it was quantitative, qualitative or mixed; the scope (descriptive or descriptive-correlational), the analysis logic used (deductive or inductive), and the time of completion of the investigation (cross-sectional or longitudinal).
9. The participants are described, with elements such as: the size of the sample, the type of people who were part of the study and their data demographics, the way of obtaining the sample, and the criteria for inclusion and exclusion.
10. The instrument or instruments used are described, indicating their authors and the data of validity and reliability that are possessed. If I dont know instruments were applied, then the collection technique of the data used in the study and how it was carried out.
11. The techniques used for data analysis are described. Collected (statistical or qualitative techniques), as well as the ethical criteria applied
12. The results are described in a systematic, organized and synthetic way, showing the most relevant and original aspects of the study, considering, as far as possible, the same order of the established purposes. They are organized from the most general to the most specific.
13. Tables and figures are used to help synthesize, contextualize, clarify or illustrate the purposes of the study. The information contained in the tables and figures is not repeated in the text.
14. Analyzes of the data are presented according to the type of study; For example, in quantitative descriptive studies it is common to use percentages, means and standard deviations, quartile analysis, mean differences, and regression analysis, *etc.*
15. A conclusion is presented for each of the purposes of the study, based on the results obtained, and this conclusion is analyzed in detail by comparing it with similar studies, which support or dispute it, with a critical analysis.
16. The most original or impactful contributions of the study, the possible practical applications of the results, and also the limitations of the research (for example, methodological difficulties, sampling deficiencies, problems with the research design, possible biases *etc*).
17. Recommendations are provided for future studies, considering the experience of the research carried out, and taking into account trends in the area. As far as possible, this should be based on arguments and considering the proposals of other authors.
18. All the references cited in the text are described, according to the APA style in its latest edition, or considering the standards of the journal in which the article is expected to be published. All references have DOI, or, failing that, the link where they can be downloaded.

19. Appendices are presented when the type of study requires it, through extra information at the end of the article or through complementary files to the text (uploaded to the journal or in the form of links to external web pages). The information that goes in the appendices has not been published.

20. The article follows the APA standards in its latest edition, or the standards of the journal in which it is expected to be published. In addition, it complies with the grammatical rules of the Spanish language. The writing is in an impersonal way, attractive from the beginning to the end, turning everything around the purposes of the study. Each paragraph is argumentative and consists of at least seven lines.

21. The format is in accordance with the standards of the journal, whose considerations may be, in general terms: space and a half line spacing, 2.54 cm margins, Times New Roman 12 font and continuous numbering at the top right. The names of the authors or information that could identify them do not appear in the text.

and affection predict self-determined motivation (*Huffman et al., 2018*; *Laukkanen, Sääkslahti & Aunola, 2020*). This means that children wish to participate in PA by their own. However, coercive parental control and lack of structure do not help to configure self-determined motivations (*Laukkanen, Sääkslahti & Aunola, 2020*). Therefore, children will participate in PA when they feel coerced by external pressures.

## Parental involvement in active lifestyles

*Alia et al. (2013)* and *Yazeedi et al. (2021)* found that one form of direct parental involvement in children's active lifestyles could be through parental modeling. More specifically, *Yazeedi et al. (2021)* verified that parents who reduced network consumption (modeling) made it easier for their children to adopt the same behavior. On the other hand, *Lev et al. (2020)* highlight co-participation with children in PA as a form of direct parental involvement. *Lev et al. (2020)* found that parents who exercised with their children helped perceive greater enjoyment during PA and, consequently, the children's participation in it increased. *Verhees et al. (2020)* found another way of involving parents in their children's active lifestyles: indirect involvement. The indirect parental involvement in the PA of their children is carried out through the preparation of tip sheets or support bulletins and causes an increase in participation in it *Verhees et al. (2020)*. This last type of parental involvement allows parents with an intense working day to be aware of their children's habits.

## Influence of parenting styles

In relation to the influence of PS on the practice of PA, *Huffman et al. (2018)* found that democratic PS was not significantly associated with moderate to vigorous physical activity levels (MVPA). However, democratic PS was positively related to the practice of light physical activity (LPA) (*Huffman et al., 2018*). *Saunders et al. (2012)* discovered a negative association between democratic PS and active means of transport (bicycle). Around the authoritarian PS, *Yaffe (2018)* claimed that this was negatively associated with the practice of PA. *Langer et al. (2014)* verified that authoritarian PS did not predict the practice of moderate-vigorous PA. *Saunders et al. (2012)* found that the authoritarian PS was slightly but positively associated with participation in sports. As regards the permissive PS, *Langer et al. (2014)* revealed that these parents allow their children to engage in non-active play in unstructured time, which does not facilitate good levels of PA practice. However, these same researchers found that in their sample, although in a non-significant way, permissive parents were mostly related to moderate-vigorous PA in their children. *Sterret et al. (2013)* postulated that permissive PS was linked to maladaptive behaviors toward health. Finally,

**Table 2 Information of the articles examined in the systematic review.**

| Data base | First author | Year | Aim | Sample characteristics | Material | Analysis methodology | More relevant results |
|-----------|--------------|------|-----|------------------------|----------|----------------------|------------------------|
| PubMed | Alia | 2013 | To examine the interaction between parental limit setting on sedentary behaviors and health factors (weight status, physical activity, fruit and vegetable intake) in African American adolescents recruited from two rural counties in the Southeastern part of the United States | $N = 67$ parent–adolescent dyads; 59% men and 61% women | Weight and height measures were assessed using standardized protocols with a Seca 880 digital scale and Shorr height board. | Data was analyzed using Statistical Package for Social Sciences (SPSS) version 17.0 and the Statistical Analysis System (SAS) software version 9.0. | No significant association was found between limit setting and fruit/vegetable consumption and PA. |
| | | | | $(Mage = 12.67;$ $SD = 5.39)$ from United States | Parent and adolescent BMI values were computed as: weight (kg)/height $(m)^2$ | Pearson product-moment correlations were used to assess multicollinearity. | Among obese parents, setting higher (vs. lower) limits is associated with lower adolescent BMI. |
| | | | | | Parental strategies for feeding and activity were evaluated through: The limiting-Activity subscale of the Parenting strategies for Eating and Activity Scale (PEAS). | Two hierarchical regressions were performed to assess the interaction of parental factors and limit setting of sedentary behaviors in adolescents. | Parental limits implemented by both parents can prevent obesity |
| | | | | | Parent fruit and vegetable intake was assessed using a fruit and vegetable screening tool. | A separate regression was run for parental physical activity. | |
| | | | | | Parent moderate-to-vigorous physical activity was assessed using the International Physical Activity Questionnaire short form (IPAQ). | | |
| Web of science | Huffman | 2018 | To examine associations between motivation, parenting factors associated with Self-Determination Theory's psychological needs and adolescent moderate to vigorous physical activity | $N = 148$ African-American adolescents (44% men and 66% women) | Parenting styles were measured with the Authoritative Parenting Index (API) | Separate hierarchical regression analyzes were used to examine the associations between the authority of the parenting style, emotional support for (PA, tangible support for PA, autonomy support for PA, PA parent modeling and adolescent motivation with moderate-to-vigorous PA (MVPA) and light PA (LPA). | PS were not associated with moderate-vigorous PA. |
| | | | | $(Mage = 13.6$ years, $SD = 1.74)$ and their parents ($Mage$ 43.4 years, $SD = 8.21)$ | Autonomy supportive parenting for physical activity (PA) was measured using a adolescent-report scale which measured shared decision making around PA and Parent emotional support for PA | | Autonomy support was related to low moderate-vigorous PA. |
| | | | | | Emotional support for PA was measured using a modified version of an instrument that assesses family support for exercise behaviors; The Support for Exercises Scales (SES)- | | The motivation for PA is strongly associated with MVPA. |
| | | | | | Adolescents' motivation for PA was measured using The Motivational Scale | | |
| | | | | | Objective measurement of PA duration and intensity was obtained using Actical omni-directional accelerometer estimates | | |

| Data base | First author | Year | Aim | Sample characteristics | Material | Analysis methodology | More relevant results |
|---|---|---|---|---|---|---|---|
| Web of science | Langer | 2014 | To examine relationships between parenting styles and practices and child moderate-to-vigorous physical activity (MVPA) and screen time | $N$ = 42 children (50.6% men and 49.4% women) ($M$age 6.9 years; $SD$= 1.8) of Minneapolis–Saint Pau (Midwestern United States) | The PS were examined through the Parenting Styles and Dimensions Questionnaire (PSDQ)<br><br>Infant PA was assessed using accelerometry (ActiGraph GT1M accelerometer).<br><br>Four items assessed the amount of time children spent on an average weekday and an average weekend day (a) watching TV and (b) playing video or computer games or using a computer for purposes other than schoolwork | Analyses were conducted using Statistical Package for the Social Sciences 20.0 and SAS 9.2.<br><br>Linear and logistic regression models were carried out to understand the connection between the variables. | Support for PA from democratic and permissive parents positively predicted moderate-vigorous PA.<br><br>Permissive and authoritarian parents positively predicted high screen consumption.<br><br>Interventions aimed at increasing PA and reducing screen time in children may benefit from including the figure of parents. |
| ResearchGate | Laukkanen (2020) | | To examine children's perspectives on parents practices and how they relate to their motivational regulation of PA | $N$ = 79 children<br><br>(48.1% girls and 51.9% boys) from Finland (Europe) | Interview | A qualitative content analysis was used, in which the research data was intended to expand the existing theoretical knowledge.<br><br>The data were then coded using qualitative analysis software (ATLAS.ti version 7.5). Finally, statistical SPSS was used to perform descriptive analyses. | Parents are defined by a higher dimension (high responsiveness and low demand). Within it, the following must be evaluated: support for autonomy, involvement, and structure. Autonomy support is related to autonomous motivations, involvement to autonomous and controlled motivations, and structure to autonomous motivations. Other parents are defined by the higher dimension: high responsiveness and high demand. In them, the structure must be considered, which is related to autonomous motivations. Other parents are defined by the higher dimension: low response and high demand. Coercive control is related to controlled motivation in this type of parents. Finally, there are parents whose higher dimension is low response and low demand. In this group, the lack of structure related to amotivation, should be valued. |

| Data base | First author | Year | Aim | Sample characteristics | Material | Analysis methodology | More relevant results |
|---|---|---|---|---|---|---|---|
| Scopus | Lev | 2020 | To shed light on parents' level of involvement with their child's sporting activity in Israel. | 173 parents (51.4% men and 48.6% women) ($Mage$ = 45.9; $SD$ = 6.4 years). | Questionnaire for parents and questionnaire for children composed of 22 questions that investigated the logistical and emotional participation of parents and the satisfaction of their children. | Statistical analysis was performed using SPSS (v.24). | Actions such as taking children to training practices, parents' participation in their children's basketball activity, and parents' interaction with their children during the game were positively correlated with child satisfaction. the children |
| | | | | 173 children, (78% men and 22% women) ($Mage$ = 13.7; $SD$ = 1.7 years) form Israel | | Student test was used to compare between the results of mothers and fathers. | Parents' emotional involvement was the most important variable for child satisfaction. |
| | | | | | | Pearson's correlation analysis was used to determine the relationship between parents' and child's answers. | |
| | | | | | | Multiple regression was used to identify which kind of involvement is the best for describing the variation in child's satisfaction | |
| PubMed | Saunders | 2012 | Described cross-sectional and longitudinal associations between parenting style and girls' participation in organized sports, walking or cycling, and engaging in vigorous physical activity (MVPA). | $N$ = 222 Adolescent girls (9–12 years) and their parents form Melbourne (Australia) | The PS were evaluated from 22 items adapted from the Parent Attitude Inquiry | Bivariable linear regression models were generated to assess associations between independent (parenting style) and dependent (organized sport, MVPA and walking/cycling trips respectively) variables. | There are positive associations between authoritarian PS and the frequency of organized sport. |
| | | | | | Participation in organized sport was self-reported using an adaptation of the Adolescent Physical Activity Recall Questionnaire | | There are negative associations between authoritarian and permissive PE and the number of trips on foot and by bicycle. |
| | | | | | Girls were asked to report how often they walked or biked to 15 common destinations. | | There are no significant associations between PE and moderate-vigorous PA. |
| | | | | | MVPA was assessed using accelerometers | | |
| Google scholar | Sterret | 2013 | To examine the relationships between 2 parenting styles and family nutrition and physical activity. | 175 parents of 26–66 years, (43 men, 9% and 132 women, 91%) ($Mage$ = 37.6; $SD$ = 6.6) from southeastern United States | The Family Nutrition and Physical Activity (FNPA) scale served as a measure to evaluate nutrition and BP at the family level. | Linear regression was performed to predict family nutrition and PA as a function of PS. | Democratic parents were associated with better PA and family nutrition (although not significantly). |
| | | | | | | | Permissive parents were associated with worse PA and family nutrition (significantly). |
| PubMed | Verhees | 2020 | To involve parents in a school-based intervention by challenging primary school children to perform physical activity and nutrition-related activities with their parents. | 226 children 100 men, 44.2% and 55.8% women) ($Mage$ 10.9; $SD$ = 1.0) from Western ethnicity (Europe) | Visual self-report instrument developed for primary school children in which they indicate food preferences and PA (pair comparison). | Linear regression between the variables examined. | Parents participated more in nutrition challenges when their children were young because, to learn to make healthy recipes, more parental help is needed. Therefore, children's dietary behavior may relate more to parents than PA behavior. |

(Continued)

| Data base | First author | Year | Aim | Sample characteristics | Material | Analysis methodology | More relevant results |
|---|---|---|---|---|---|---|---|
| | | | | | The parents completed the Family Health Climate scale (FHC) That measures family perceptions and cognitions about nutrition (FHC-NU) and the PA (FHC-PA) of the children. | | It is recommended that families be considered in interventions that require a change in dietary and PA behavior. |
| PubMed | Yaffie | 2018 | To examine the association between parenting styles and adolescent physical activity among Israeli Arab families of adolescent boys | $N = 177$ Israeli Arab male adolescents with normal weight ($Mage = 13.93$, $SD = 1.42$) | To measure the PS, it was used The Parental Authority Questionnaire (PAQ) | The Regression Analyses Results of Predicting Israeli-Arab Adolescents' PA from the PS | Adolescents who perceived their parents as democratic were more essentially active than those who perceived them as authoritarian. |
| | | | | | The participants' physical activity level was evaluated through a three-item questionnaire in which the frequency and intensity of PA in a particular week was investigated. | | Permissive PS are not statistically significantly related to PA. |
| PubMed | Yazeedi | 2021 | The objective of this study is to expand the understanding of the family influence on children's nutrition and physical activity patterns in Oman | $N = 204$ dyads (Asian mother with a child). Mothers ($Mage$ 28.71, $SD= 5.52$).<br><br>Children (93 men 47.4% and women 103 52.6%). $Mage = 7.74$, $SD = 5.52$) from Oman (Asia) | Family nutrition and physical activity patterns were assessed with the Arabic version of the Family Nutrition and Physical Activity (FNPA BAR) | Bivariate and multivariate analysis, including correlation, independent t-test and chi-square, Multiple linear regression and binary logistic regression were used to analyze the relationships between the study variables. | Moderate-vigorous PA was not related to parent ratings on the FNPA. |

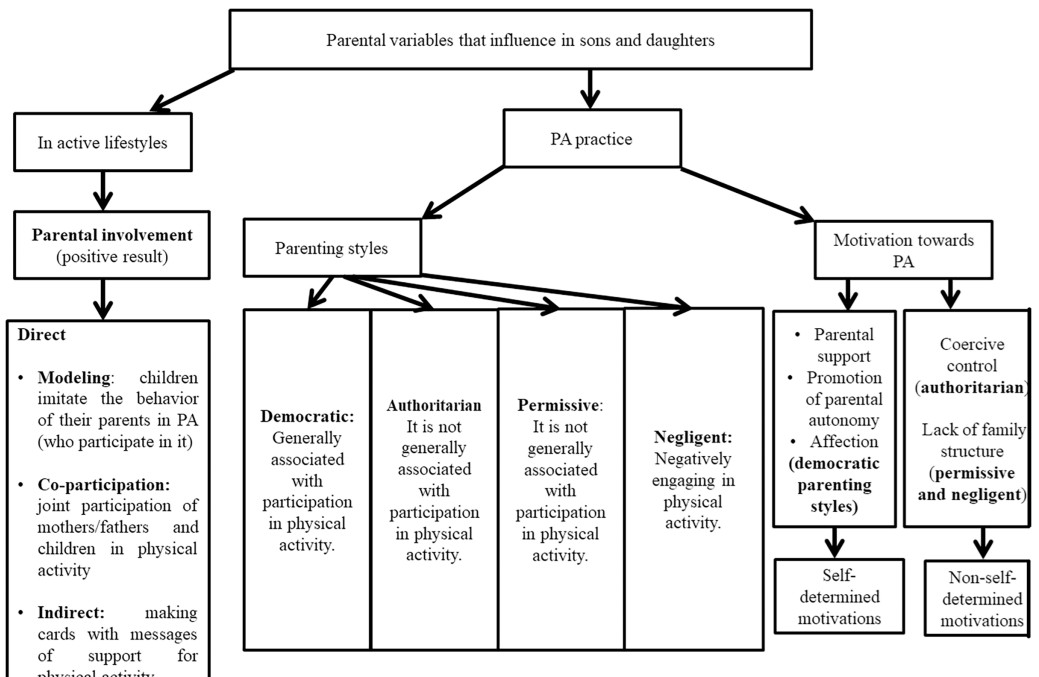

**Figure 2 Summary diagram of the results obtained in the variables of parental involvement and parenting styles in active lifestyles and motivation towards physical activity.**

**Table 3 Parental variables that influence active lifestyles and the motivation towards physical activity.**

| Positive influence | Negative influence |
| --- | --- |
| – Parental modeling based on active lifestyles | – Sedentary parental modeling |
| – Parenting support and affection | – Authoritarianism and excessive permissiveness |
| – Co-participation in physical activity with children | – Lack of support and affection |
| – Observation | – Deprivation of supervisión |

regarding neglectful PS, *Saunders et al. (2012)* found a negative association between the same and organized sports practice.

Regarding the influence of PS on motivation, it was found that parental support was positively correlated with motivation toward PA (*Huffman et al., 2018*). On the other hand, *Laukkanen, Sääkslahti & Aunola (2020)* verified that parents who promoted autonomy in their children's PA facilitated their perceptions of independence and helped them originate autonomous motivation (self-determined). Likewise, the children who perceived support, warmth and their parents demanded reasonable expectations also contributed to the configuration of self-determined motivations in PA. In line with what was described, the children who perceived demanding and strict parental expectations towards PA seemed to identify them as signs of support towards perceived competence and originated self-determined motivations (*Laukkanen, Sääkslahti & Aunola, 2020*).

However, according to *Laukkanen, Sääkslahti & Aunola (2020)* coercive parental control encouraged children to perceive autonomy dissatisfaction and generated non-self-determined motivations. *Laukkanen, Sääkslahti & Aunola (2020)* also verified that the lack of parental structure created low self-determined motivation in PA.

## DISCUSSION

This systematic review aimed to know the influence of parental involvement in children's active lifestyles and the influence of parenting styles on children's practice and motivation towards physical activity. This review conveys the importance of parents' involvement in children's active lifestyles through parental modeling in early childhood and co-participation in PA. On the other hand, this work also reflects the influence of PS on PA. In this case, it can be seen how maternal and paternal support does not always provide good results in the practice of PA. This is verified in some studies where there was no relationship between both variables or where the absence of parental support was related to the practice of PA. Finally, this review reveals that parental support is related to strong self-determined motivation towards PA, same as affection, appropriate demanding requirements and non-coercive control.

Regarding the influence of parental involvement in active lifestyles, it is considered that parents can promote the aforementioned active lifestyles thanks to alternatives such as direct involvement with adequate parental modeling (*Alia et al., 2013*; *Yazeedi et al., 2021*), and participation together with the children in the exercise (*Lev et al., 2020*). The positive effect of parental modeling found in the work of *Alia et al. (2013)* and *Yazeedi et al. (2021)* is consistent with other projects, where it was found that children tended to be active when their parents were too (*Beets, Cardinal & Alderman, 2010*; *Nuviala, Ruíz & García, 2003*; *Ormrod, 1999*). Considering the joint participation of parents and children in PA, *Alia et al. (2013)* and *Yazeedi et al. (2021)* found a positive effect on children's PA practice mediated by parent-child co-participation. These results are consistent with what was stipulated by *Lev et al. (2020)*, who establish that parents who exercise with their children encourage them to perceive greater enjoyment during PA (and increase their participation in it). *Verhees et al. (2020)* allude to another alternative to parental involvement, which is indirect involvement. This can be done by preparing sheets or support newsletters from mothers and fathers. This alternative allows parents who have an intense working day to be involved in the habits of their children (*Verhees et al., 2020*).

Concerning the influence of PS in PA, a brief allusion is made to the different styles: warm and controlling parents (democratic), non-affective but controlling (authoritarian), affective but not controlling (permissive) and non-affective and non-controlling (negligent). *Huffman et al. (2018)* found that democratic PS was not significantly associated with moderate-vigorous physical activity (but it was related to light physical activity). According to *Huffman et al. (2018)* these results may be influenced because they worked with an overweight sample, who tend to perform fewer minutes of PA compared to healthy-weight active youth. The results of *Huffman et al. (2018)* are consistent with those of *Langer et al. (2014)*, who did not find a relationship between democratic style and the practice of the aforementioned moderate-vigorous physical activity. In the study by *Langer*

*et al. (2014)* seemed to prove that in PA when parents had little permissiveness, support did not matter. In the case of democratic parents, support is a variable that is present, however, there is also the control variable (*González-García, 2017*). Perhaps for this reason, no significant relationship was found between democratic PS and moderate-vigorous physical activity. Regarding these results, previous studies consider that the variables of reasoning and accepting children's requests in PA (present in democratic style) facilitate participation in PA (*Hennessy et al., 2010*). Therefore, these variables may have a greater influence on the predisposition to participate in light physical activity than the moderate-vigorous PA. On the other hand, *Saunders et al. (2012)* found that the authoritarian style was slightly but positively associated with participation in sports. *Saunders et al. (2012)* specify that parental control can prevent certain behaviors in children (such as accepting minors' requests to be taken on inactive means of transportation when traveling). Perhaps this "control" is responsible for their increased participation in sports. In another previous work, *Jago et al. (2011)* described how some variables related to authoritarian PS were related to low levels of PA practice, such as over-demands and impositions. Considering the heterogeneity of results, it can be verified that authoritarian PS are not always synonymous with physical inactivity. Around the permissive PS, *Langer et al. (2014)* could not detect any association between this style and the practice of PA. In their work, these researchers show that when parents were very permissive, high support was protector and scarce support were detrimental. *A priori*, in this sample, the high degree of permissiveness could be accompanied by little support for PA, which could lead participants to choose sedentary activities over the practice of PA. On the other hand, *Sterret et al. (2013)* verified that children raised by permissive PS were related to low levels of PA. This occurred because excess permissiveness was related to maladaptive behaviors regarding physical health (*Sterret et al., 2013*). Previously (*Torio, 2008*) showed that permissive parents do not control their children's behaviors during parenting (*Torio, 2008*). Likewise, *Mazlina, Atikah & Rozile (2019)* mentioned that parental supervision during PA practice helps children's participation. Given that this variable tends to be non-existent in permissive PS, this could explain the negative relationship between PA and these PS (*Sterret et al., 2013*). Finally, *Saunders et al. (2012)* described a negative association between negligent PS and the practice of sport. In their work, a trend is observed towards a longer duration of organized sport with authoritarian PS (which implies control). Both support (*Hennessy et al., 2010*; *Jago et al., 2011*) and control (*Mazlina, Atikah & Rozile, 2019*) promote participation in PA. Thus, its absence could explain the negative relationship between the negligent PS and PA.

Regarding the motivational variable, according to the influence of PS, *Huffman et al. (2018)* found that the democratic PS favored self-determined motivation. *Tang et al. (2018)* support this premise since, in their study, the democratic PS was not associated with motivations with low self-determination. On the other hand, *Laukkanen, Sääkslahti & Aunola (2020)* found that parents who promoted autonomy in their children's PA facilitated their perceptions of independence and helped them originate autonomous motivation (self-determined). Namely, *Laukkanen, Sääkslahti & Aunola (2020)* affirm that children who perceived support, warmth and their parents demanded them reasonable

expectations, configure self-determined (autonomous) motivations. Both the promotion of autonomy and the support, warmth and adequate demands of exigency are variables related to the democratic PS (*Torio, 2008*). Therefore, once again, the close connection between democratic PS and self-determined motivations toward PA seems to be verified (*Huffman et al., 2018*; *Laukkanen, Sääkslahti & Aunola, 2020*). On the other hand, *Laukkanen, Sääkslahti & Aunola (2020)* found in their study that children who perceived demanding and strict parental expectations (authoritarian PS) towards PA developed self-determined motivations toward the same. Previously, authoritarian PS was negatively related to self-determined motivation (*Tang et al., 2018*). However, in this case, the children understood the over-demand as a variable of parental support and that it was aimed at improving their perception of competence during the PA (*Laukkanen, Sääkslahti & Aunola, 2020*), so it does not weaken their self-determined motivation. Another variable that can appear in the authoritarian PS is coercive control. In the study of *Laukkanen, Sääkslahti & Aunola (2020)*, it seems to be responsible for creating non-self-determined motivations. This happens because it deprives the perception of autonomy during PA. Finally, the lack of parental structure fostered sedentary behaviors in the children, a low perception of competence during PA, and created low self-determined motivation (*Laukkanen, Sääkslahti & Aunola, 2020*). This result is in common with what was exposed by *Tang et al. (2018)*, who argued that permissive PS does not create self-determined motivations toward PA.

## Limitations, lines of future research, and practical implications

A limitation of this work is that the variables analyzed in the different research were not examined with the same instruments. In addition, manuscripts that exclusively categorize PS as authoritarian, democratic, permissive, and negligent have been included, preventing other aspects related to positive parenting or other theories from being considered. In addition, recent articles have been included, considerably reducing the number of available investigations. Finally, the methodological limitations of the systematic review must be exposed. In this case, the scores provided to assess the analysis of the quality of the consulted manuscripts carry a certain degree of subjectivity. In future research, it is recommended to replicate the study considering other theories of PS, include older PS manuscripts, and establish a comparison between the results of yesteryear with the current ones to see the evolution of the impact of PS on the variables examined. The decision to choose manuscripts based on the categorization of democratic, permissive, authoritarian, and negligent style allows for obtaining more results than other PS models, since most of the instruments used in sports and PA are based on said classification. Finally, having chosen recent articles is justified because parental education fluctuates over time and the results obtained are adjusted to the reality of today's society (*Ching-Man, Wai-Man & Sin-Min, 2019*).

This systematic review is helpful to identify the variables of parental involvement (modeling, co-participation with children in PA) and PS (parental support, affection and control) that are most suitable for promoting active lifestyles. Thanks to the promotion of active lifestyles, diseases such as obesity, cancer, metabolic syndrome, and cardiovascular

diseases, which are associated with low levels of PA can be avoided (*Guo et al., 2019*; *Lätt et al., 2015*). In addition, this research is feasible to reveal how families may be responsible for the perception of feelings of unhappiness during children's PA practice. This can happen when authoritarian PS impose excessive sports demands on their children or do not allow them to choose the PA they want to do. Finally, this work is crucial to reveal that parental support is important to create self-determined motivations towards PA and, therefore, to encourage healthy behaviors to last over time.

## CONCLUSIONS

In conclusion, parents can act as role models (doing PA) for their children to imitate or exercise simultaneously with them. This direct parental involvement will reduce sedentary children's behavior associated with the current lifestyle. Whenever there are difficulties in reconciling family life with work, parents should use indirect involvement to encourage their children's active lifestyles through messages or letters of support. Again, this involvement will translate into an active lifestyle. Parents should support their children in the practice of PA because, in this way, they facilitate their interest in participating in it, and promote enjoyment during exercise. Support, together with the variables of affection, appropriate demanding requirements, and non-coercive control, helps children form self-determined motivations for active lifestyles, ensuring that their behaviors are not lost over time.

### Funding
The authors received no funding for this work.

### Competing Interests
The authors declare that they have no competing interests.

### Author Contributions
- Marta Vega-Díaz conceived and designed the experiments, performed the experiments, analyzed the data, prepared figures and/or tables, authored or reviewed drafts of the article, and approved the final draft.
- Higinio González-García conceived and designed the experiments, performed the experiments, analyzed the data, prepared figures and/or tables, authored or reviewed drafts of the article, and approved the final draft.
- Carmen de Labra conceived and designed the experiments, performed the experiments, analyzed the data, prepared figures and/or tables, authored or reviewed drafts of the article, and approved the final draft.

## Data Availability

This is a systematic review.

## Supplemental Information

Supplemental information for this article can be found online at http://dx.doi.org/10.7717/peerj.16668#supplemental-information.

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
