# Peer review of "Influence of parental involvement and parenting styles in children’s active lifestyle: a systematic review"

_PeerJ, doi:10.7717/peerj.16668_

## Round 0.1 · original submission · Major Revisions

Please read all the comments and suggestions carefully and revise them accordingly. They are all valuable and constructive. I cannot foretell whether your manuscript will be published or not. It all depends on how appropriately you revise it.

·

Basic reporting

What did past systematic review articles address? And why did author/s conduct another systematic review ? In addition, since there have been separate systematic reviews, why should these two separate systematic studies be merged and discussed together? We have seen that the author/s have conducted comprehensive discussions in the discussion chapter, but it is recommended that the introduction should further explain the reasons for conducting another systematic review and conducting a systematic review of the two together.

Experimental design

1.The PRISMA statement has been updated to the 2020 version. It is recommended to correct it according to this version. (The PRISMA 2020 statement: An updated guideline for reporting systematic reviews/ https://doi.org/10.1016/j.ijsu.2021.105906)

2.Based on the quality assessment, it is not stated in the article whether it is possible to specify what score is required to indicate that this article is of high quality.

Validity of the findings

no comment

·

Basic reporting

introduction needs more adequately introduce the subject, and please indicate the detailed information of "active life" and the definition of it. It seems you used “PA” instead of “active life," but the study you included seems active life also covers screen time and eating habits. The topic is "active life," but the article only stated "PA." Please make it by consistent.

6.You should indicate what level of PA represents an active lifestyle.
7.How can we prove that "59.6% of the Spanish population practiced regular sports..." means "This data shows that the prevalence of a sedentary lifestyle.... is very high”? What is the global average? What percentage is high? What percentage is low? Is there any past research to support this conclusion?
8.Line 79. Is there any literature support for the above? Please note the references.
9.Line 87-89. Instead of writing about the purpose of the study, I suggest that the effect of parental involvement or lifestyle on children's participation in physical activity should be described as a bridge between the previous and the following.
10.Line 99-100. Unclear expression: what you want to express is " 76% of young people are physically active, whose mothers actively practiced sports"?
11.Line 111-116. It is suggested that this paragraph should be combined with the next section to explain the effect or influence of different types of PS on children's PA.

Experimental design

Inclusion criteria need to be clarified. Not giving enough details, such as "What kind of study design?" "What kind of outcomes? " might include. And what does " study the levels of PA practice" mean? Where is the research object of the child presented? Please describe how the study selection was conducted/ how the study was screened.
14.Line 189: the word “EASH” seems incorrect.
15.After reading the quality assessment, I still needed clarification on the scale. What is the total score on the scale? How many rating levels are there?

Validity of the findings

Result
16.Line 210-211. Please describe the subject clearly.
17.Line 236. The word "state" seems incorrect; please check English tenses in full text.

Discussion
18.Line 261-262. The subject needs to be clarified. The formulation misses the point and seems irrelevant to the topic.
19.Line 264. “from” seems incorrect.
20.Line 264-265. “(because....sedentary lifestyle)” This sentence does not seem necessary to describe here.
21.The result found that democratic PS was positively related to the LPA, but others found it is negative with active means of transport. The discussion section should explain this different point of contention. Point out the reasons for the difference rather than simply listing the same points as other studies.
22.Like the "authoritarian style," you found different research results but did not explain or discuss this contention in the discussion.
23.Line 307. “Since this variable will not be present," Line 308 "Finally," the twist seems inadequate.
24.Line 355-356. “Those variables will prevent ...metabolic syndrome. " This is abrupt and needs a reason to connect the statements.
25.Limitations and future suggestions, or strengths of the study, should separate a paragraph with the subheading.
26.Line 366-367. Again, associated with "whose" current lifestyle? Many similar statements in the article need to be clarified, and it needs to be clarified whether the type of parent influences the parents' own behavior or the influence of the child, often without identifying the child as the object.

Additional comments

Abstract
1.The purpose of the study needs to be articulated more clearly. I suggest you improve the description in line 37-38 to clarify the purpose of the research.
2.Line 39. Missing punctuation endings.
3.Line 40. The methodology should briefly describe the methods used, such as the guidelines, specific quality assessment methods, etc.
4.Line 43. People want to know from the results in the abstract how past studies have found that parental involvement and parenting styles impact children's positive lives rather than simply categorizing the literature results. I recommend adding specific effects from the analysis of all included literature.

Table
27.Table 1: The last column of Table 1 is not clear.
28.If the categorized PS is used primarily, it should be presented as a separate column in Table 2.

·

Basic reporting

This article tried to find the influence of parental involvement and parenting style on children's active lifestyles with a systematic review method. Through this article, readers could understand how important it is for parents to be positive models and their parenting styles to facilitate children's physical activity. Nowadays, this issue is truly critical. However, the work needs to be improved in some respects, some of them important.
1. The title: "active life" is different from "active lifestyle"; according to the context, it seems to focus more on the lifestyle. Authors could consider the term to make it more concrete.
2. In the introduction starts with the influence of health by inadequate PA, and people should live an active lifestyle from their childhood in the first paragraph. But the next paragraph starts from the aim of this study; it seems to lack the reasons or gap why parental involvement and parenting style … were so important, and this issue is worth exploring.

Experimental design

1. The reason for the period of article searching is "because PS vary over time…" (line 168-169). Is there any evidence or reference showing that PS was so different between the last decade (2012-2022) and earlier decade (2002-2011) that this work needn't adopt earlier articles?
2. compare the methodology and abstract. The number of articles on the first screen is 951. Or 997 (line 178)? Furthermore, according to Figure 1, there are "46 articles obtained from additional works through other sources and references." I recommend indicating how they have been found.
3. Line 186-194. The tool used for assessing the quality of included articles is "the Scale for(or to?) Evaluating Scientific Articles in Social Human Science." the abbreviation of this tool should use the English version (SSAHS) so that the reader can understand that. And the acronym in this paragraph should be consistent.

Validity of the findings

1. Line 199-201. The included articles are 10, but these three sentences indicated "one of the nine…" and "four of the nine…" which seems only nine articles were included. Why the other article was not mentioned here ( Steerret, 2013; 175 parents)?
2. In the results, what measurements were used to measure the parental involvement, parenting styles, and physical activity…, so readers could understand these variables more?

Additional comments

1. In Table 2: "The more relevant results" could add the main results related to the main variables of this systematic review article (PI, PS, PA, & SDM); it will help readers understand and contrast the main results among included articles. Ex: lines 232-233, the democratic PS and active means of transport have a negative association and would not be easy to compare with the context of Table 2.
2. In Table 2, the column of "sample characteristics" of Laukkanen's article (2020) describes another "48.10% of women and 51.90% of men". Does it mean anything?
3. In Figure 1, it needs to be clearer to read it.

---

## Round 0.2 · Minor Revisions

The reviewers give several comments and suggestions after your second version. I hope you revise it accordingly. Then, I will make my decision.

·

Basic reporting

Abstract

1. Line 43-44
“Furthermore, it was utilized the Preferred Reporting Items for Systematic Reviews and Meta-Analyses 2020. ”- I suggest providing the explanation in the "Quality Assessment" section.

2. Line45-51
A total of 10 articles were included in this study. Based on the results and Table 2, some articles showed significant findings while others did not. It is recommended that each of these 10 articles should provide a brief summary to support the "conclusion" statement.

Experimental design

Method

Line 180-205
It is recommended to present information in a coherent sentence format rather than using bullet points. This approach enhances the smoothness and coherence of the text. The paragraph primarily contains information about the number of researchers involved in literature search, the handling of discrepancies in the number of included articles when searching independently, the number of articles excluded at each stage of the search process, and the reasons for exclusion.

Line 199-205
The content should be consistent with Figure 1. In summary, during the retrieval process, how many duplicate articles were excluded through database or other index searches (the author/s' first version was 46 articles)? How many articles were excluded during the abstract and title screening step? And how many articles were excluded during the full-text screening step? (What were the reasons for exclusion?) This provides guidance for the necessary revisions.

Validity of the findings

Result

Line 237
“However, in two of the nine works……….” Is it nine or ten?

Additional comments

Figure1

1. According to the PRISMA 2020 flow diagram template for systematic reviews, the author/s' first version was to distinguish “Identification of new studies via databases and registers” and “Identification of new studies via other methods”, so the flow chart should be produced from these two parts.
2. “Duplicate records removed” should be explained during the searching step of “Identification of new studies via databases and registers” and “Identification of new studies via other methods”. In addition, the author/s' finally presented 31 articles, but only ten were included in this study.
3. It is recommended to read the 2020 version carefully (The PRISMA 2020 statement: An updated guideline for reporting systematic reviews/ https://doi.org/10.1016/j.ijsu.2021.105906/ Figure1)

·

Basic reporting

Abstract

The methodology should also mention the duration of the searched articles (the publication date).

Experimental design

The author/s explained why PS varied over time, especially the difference between the last decade and earlier decades. However, the reasons should mentioned in this study so that the reader can understand the reasons why the author/s only searched the articles between 2012 and 2022.

Validity of the findings

The included articles were 10, but in Line 237, “ in two of the nine works,...” is that correct?

Line 282~283, “Coercive parental control encourages children to perceive autonomy dissatisfaction and generated non-self-determined motivations.” Do this description according to any other reference.

Additional comments

In Figure 1, the total number of studies included in the review should be 10, not 31 articles.

---

## Round 0.3 · accepted · Accept

Please check your reference. There are many cited references with underlines. Please delete it.